# Mesenchymal Stromal Cell-Based Targeted Therapy Pancreatic Cancer: Progress and Challenges

**DOI:** 10.3390/ijms24043559

**Published:** 2023-02-10

**Authors:** Zhilong Ma, Jie Hua, Jiang Liu, Bo Zhang, Wei Wang, Xianjun Yu, Jin Xu

**Affiliations:** 1Department of Pancreatic Surgery, Fudan University Shanghai Cancer Center, No. 270 Dong’An Road, Shanghai 200032, China; 2Department of Oncology, Shanghai Medical College, Fudan University, Shanghai 200032, China; 3Shanghai Pancreatic Cancer Institute, No. 270 Dong’An Road, Shanghai 200032, China; 4Pancreatic Cancer Institute, Fudan University, Shanghai 200032, China

**Keywords:** pancreatic cancer, mesenchymal stromal cells, exosomes, tumor-targeted therapy

## Abstract

Pancreatic cancer is an aggressive malignancy with high mortality rates and poor prognoses. Despite rapid progress in the diagnosis and treatment of pancreatic cancer, the efficacy of current therapeutic strategies remains limited. Hence, better alternative therapeutic options for treating pancreatic cancer need to be urgently explored. Mesenchymal stromal cells (MSCs) have recently received much attention as a potential therapy for pancreatic cancer owing to their tumor-homing properties. However, the specific antitumor effect of MSCs is still controversial. To this end, we aimed to focus on the potential anti-cancer treatment prospects of the MSC-based approach and summarize current challenges in the clinical application of MSCs to treat pancreatic cancer.

## 1. Introduction

Pancreatic cancer, an aggressive human malignant tumor, is often termed a silent killer owing to its poor prognosis, and its incidence has been increasing over the years [1,2,3]. The mortality rate within one year after diagnosis is approximately 75%, and the 5-year survival rate is no more than 8% [4,5,6]. Pancreatic ductal adenocarcinoma (PDAC) accounts for 90% of all pancreatic tumors, and other subtypes include acinar carcinoma, pancreatoblastoma, and neuroendocrine neoplasms [3]. Approximately 50% of patients with PDAC display no symptoms during the early stage, and by the time a diagnosis is confirmed, they are in the late stage of PDAC [3,7,8]. Thus, most patients miss opportunities for radical surgical resection in the early stage and can only undergo radiotherapy and chemotherapy later. However, owing to the special extracellular matrix barrier of pancreatic cancer and resistance to chemotherapeutic drugs, some cancer cells cannot be killed [9,10]. Furthermore, approximately 40% of patients with PDAC experience tumor recurrence even after surgical resection and die within one year [11,12,13]. Despite the rapid progress in the diagnosis and therapy of pancreatic cancer, the efficacy of present therapeutic measures remains poor [7]. Therefore, identifying alternative treatment strategies for the better management of pancreatic cancer is an urgent requirement.

As a carrier of anti-tumor drugs, mesenchymal stromal cells (MSCs) can be genetically engineered to release various agents such as treatment proteins, suicide genes, and oncolytic viruses to decrease cancer growth and progression [14,15]. The application of MSCs as therapeutic biological carriers in cytotherapy has some distinct advantages, including low immunogenicity, tumor tropism, a massive expansion in vitro, and the ability to transfer various therapeutic agents [16,17,18,19]. Reportedly, MSCs can home to tumor locations and survive in the tumor microenvironment (TME) [16,20,21]. Recent research has indicated that MSCs could be used as a promising “weapon” in treating pancreatic cancer [22,23,24].

Despite the therapeutic potential of MSCs in several diseases, their applications in the treatment of cancer are still controversial. Although MSCs have been found to promote tumor initiation and progression, emerging research has indicated the beneficial effects of MSCs in cancer therapy [17,25,26,27]. In this review, we aimed to focus on the potential anti-cancer treatment prospects of the MSC-based approach and summarize current challenges in the clinical application of MSCs for treating pancreatic cancer.

## 2. Treatments and Challenges of Pancreatic Cancer

Based on the extent of the tumor, patients are often divided into four categories, namely those with resectable, borderline resectable, locally advanced, and metastatic tumors. Although the 5-year survival rate of patients who can undergo surgical resection is 10–25%, surgery remains the only curative intervention [7,28,29]. Adjuvant chemotherapy, namely the modified FOLFIRINOX (fluorouracil, oxaliplatin, irinotecan, leucovorin) and gemcitabine plus capecitabine or gemcitabine alone, is recommended after PDAC resection for patients with high and poor functional status, respectively [3]. Additionally, neoadjuvant and perioperative treatments are recommended for resectable and borderline resectable tumors. Eradication of occult metastatic disease could increase the number of patients eligible for systematic treatment. Despite the controversial role of radiotherapy in localized PDAC, the present guidelines support neoadjuvant chemotherapy with or without radiotherapy for local diseases as a therapeutic intervention [3,30,31]. In approximately 80% of inoperable locally advanced PDAC cases, poor efficacy has been observed for new adjuvant therapy; therefore, the surgical resection rate cannot be increased. The modified FOLFIRINOX or albumin-bound paclitaxel and gemcitabine are used to slow down tumor progression [3,7,32]. Furthermore, the role of radiation in locally advanced PDAC is still controversial. About 50% of patients have distant metastasis at the time of diagnosis, and systemic chemotherapy continues to be the primary intervention for alleviating cancer-related symptoms and prolonging life [7]. Currently, gemcitabine and albumin-bound paclitaxel or modified FOLFIRINOX is still the standard first-line therapy for metastatic patients [3,7,33,34].

Based on current studies, the failure of chemotherapy, targeted therapy, and immunotherapy for pancreatic cancer may be attributed to the special characteristics of pancreatic cancer, including high malnutrition, immunosuppression, hypoxia, and desmoplastic characteristics [35]. Pancreatic tumor cells are surrounded by an abundant desmoplastic stroma that forms the TME and is composed of fibroblasts, pancreatic stellate cells (PSCs), MSCs, immune cells, blood vessels, and extracellular matrix proteins [36]. This abundant stroma is a physical barrier that hinders the effective delivery of chemotherapies to tumors. Furthermore, the specific TME plays key roles in tumor biology and modulates the immune recognition of pancreatic cancer cells [37,38].

The failure of conventional chemotherapy is due to chemotherapy resistance and abnormally abundant extracellular matrix [39,40]. Therapies targeting programmed death 1 or programmed death 1 ligand 1 (PD-1/PD-L1) have been rapidly developed as antitumor therapy for several cancers [41]. However, the desmoplastic TME aids the cancer cells in escaping the immune checkpoints (PD-1/PD-L1) and promotes the growth and metastasis of pancreatic cancer. Thus, current anti-PD-1/PD-L1 immunotherapy has a poor therapeutic effect on pancreatic cancer [4,42,43]. Chimeric antigen receptor (CAR)-modified T-cell therapy is being rapidly developed for various cancers; however, it has been unsuccessful in displaying a potential clinical value and improving survival in pancreatic cancer [44,45]. Owing to the physical and environmental barriers in pancreatic cancer-specific TME, the infiltration of CAR-modified T cells is limited [45], as well as emerging exhaustion and missing persistence.

Hence, to overcome the pathophysiological barrier of pancreatic cancer, several MSC-based therapy strategies have been proposed.

## 3. Tumor-Homing Properties of MSCs

MSCs are adult stem cells capable of multilineage differentiation and self-renewal [46,47]. MSCs exist in most tissues and are usually extracted from various sources, including bone marrow, umbilical cord, menstrual blood, placenta, adipose tissues, and muscles [19,48,49,50]. To date, MSCs have been shown to treat multiple diseases owing to their immunomodulatory and anti-inflammatory effects and tissue repair ability [18,51,52]. They thus have excellent application prospects in regenerative medicine.

MSCs can accurately migrate to injured tissues and organs and play a key role in inhibiting inflammation, decreasing tissue fibrosis formation, and promoting regeneration, thereby indicating that MSCs can selectively migrate to certain sites in the body [53,54]. Moreover, MSCs have been found to selectively migrate to primary and metastatic tumor locations, thus revealing the tumor-homing capacity of MSCs [16,20,21,55,56,57,58,59]. However, despite reports that MSCs could migrate to tumor locations in various types of tumors, the potential mechanisms by which MSCs home to tumors are still unclear.

MSCs express various chemokines and cell adhesion molecules that coordinate the mobilization of MSCs to the damage locations [15,60,61,62,63,64]. Recent research has found that the tumor-homing capacity of MSCs could be regulated by the cooperation of cytokines, chemokines, and adhesion molecules [15,24,65,66,67,68]. Hence, this observation indicates that the homing capabilities of MSCs could depend on the inflammatory microenvironment of the tumor.

MSCs are involved in the initiation, development, progression, and metastasis of tumors [69]. They can directly affect tumor development through crosstalk with tumors or the release of soluble molecules [14]. Although MSCs are indicated to exhibit pro-tumor effects [70], they can also inhibit the growth of tumors by various mechanisms, such as inhibiting tumor cell proliferation and promoting tumor cell death [71]. Furthermore, owing to their tumor-homing properties, MSCs and their exosomes have been considered promising tools for the accurate and selective release of antitumor molecules, RNA, or anticancer drugs that aid in reducing tumor cell viability and invasive characteristics [16,17,72,73,74]. Therefore, MSCs may serve as a potential tumor-targeted therapeutic strategy.

On infusing MSCs into rats, MSCs could home to pancreatic cancer sites to exert their anticancer effects [57,58]. However, the mechanism by which MSCs crosstalk with tumor cells has not yet been elucidated, and consequently, translational medicine progress has been limited. Thus, the underlying molecular mechanism by which MSCs crosstalk with tumors needs further exploration; this will aid in improving the effectiveness of MSC therapeutic potential.

## 4. MSC Therapy for Pancreatic Cancer

### 4.1. Native MSCs

Naïve MSCs have some potential advantages for treatment, such as abundance, low immunogenicity, and ease of isolation and in vitro expansion. Hence, it is widely applied to various diseases including cancer. Cousin et al. found that native human adipose tissue-derived mesenchymal stromal cells (AD-MSCs) inhibit pancreatic cancer cell proliferation and promote tumor cell death by inhibiting the cell cycle at the G1 phase [75]. Doi et al. observed that native rat umbilical cord matrix-derived stem cells (UCMSCs) decrease the growth of pancreatic tumors in mouse peritoneal models and increase the overall survival time of mice [76]. Furthermore, native human amniotic MSCs inhibit the proliferation of pancreatic cancer cells and promote apoptosis of pancreatic cancer cells by inhibiting the expression of EGFR, c-Src, and SGK223 [77]. However, although native MSCs can inhibit the growth and angiogenesis in various tumor cells, they might act as a double-edged sword while cross-talking and interacting with tumor cells [78]. Thus, the effectiveness of MSCs in decreasing the growth of pancreatic cancer needs further exploration.

### 4.2. Genetically Engineered MSCs

Genetically modified MSCs are promising potential cancer therapies to further enhance the efficacy of MSCs to target tumor cells. These MSCs deliver anti-proliferative, pro-apoptotic, and anti-angiogenic molecules to target tumor cells [70,79]. These effects might depend on several mechanisms: MSCs preferentially migrate to locations of inflammation, ischemia, and malignancy; genetically modified MSCs only release therapeutic gene products in the special TME; transgenes encoding biologic agents might themselves exhibit targeted and differential effects in tumor cells [80]. Previous studies have shown that selective targeting of therapeutic gene expression by MSCs is feasible and effective in the treatment of various cancers [81,82,83]. For example, MSCs were genetically modified to express gene products, including IFNs, CX3CL1, FLT3, tumor necrosis factor α-related apoptosis-inducing ligand (TRAIL), HGF, and GDEPT, with direct anti-tumor activity in various cancers, such as melanoma, glioma, and breast, lung, and prostate cancers [74,80,84]. This “Trojan horse” that combines MSCs with gene therapy represents a new therapeutic strategy for targeting and treating cancers.

TRAIL is a therapeutic protein that induces tumor cell death; however, pancreatic cancer cells present intrinsic resistance toward TRAIL by the expression of anti-apoptotic proteins like the X-linked inhibitor of apoptosis protein (XIAP) [85,86]. Inhibiting XIAP could promote TRAIL-induced apoptosis of pancreatic cancer cells [87,88]. Mohr et al. found that the TRAIL-modified mouse bone marrow MSCs (BM-MSCs) deliver soluble TRAIL that suppresses the metastatic growth of pancreatic cancers [89]. Moreover, TRAIL-transfected pancreas-derived MSCs can promote pancreatic cancer cell death [90]. Recently, Spano et al. observed that human AD-MSCs can be modified to efficiently release soluble TRAIL, inducing pancreatic cancer cell death and inhibiting the growth of PDAC [91]. Furthermore, by using photochemical internalization for enhanced transfection efficiency of secreting TRAIL from MSCs, the pancreatic tumor-homing properties of MSCs were enhanced [58]. This finding may provide a potential therapeutic strategy in pancreatic cancer gene therapy and clinical applications.

Zischek et al. found that the suicide gene herpes simplex virus thymidine kinase (HSV-TK)-engineered mouse MSCs can decrease the growth and metastasis of primary pancreatic cancer [92]. Sun et al. showed that NK4-modified rat BM-MSCs can inhibit the proliferation and migration of pancreatic cancer cells [93]. Schug et al. found that NIS-modified MSCs combined with 131I application can inhibit the growth of pancreatic cancer cells [94]. Notably, in a pancreatic cancer model, Tie2/TK-engineered MSCs can significantly decrease primary tumor growth [95]. Moreover, IFN-β-engineered BM-MSCs can selectively home to locations of primary and metastatic pancreatic cancers, regulating TME and reducing the growth of pancreatic cancer [96]. IL-15-engineered UCMSCs can specifically home to pancreatic cancer sites, and a sufficient amount of UCMSC-IL15 has been observed to survive to release IL-15, which could significantly suppress the growth of pancreatic cancer [97]. Furthermore, IL-10-engineered BM-MSCs can decrease the growth of pancreatic cancer by reducing the expression of IL-6 and TNF-a and inhibiting tumor angiogenesis [98].

In summary, combining MSCs with selective gene treatment results in enhanced therapeutic effects on inhibiting tumor growth; this might aid in developing new tools for pancreatic cancer treatment (Figure 1). 

### 4.3. Exosomes as a Vehicle for Therapy Delivery

Thus far, numerous studies have found that MSCs can secrete extracellular vesicles), including microvesicles (100–2000 nm in diameter) and exosomes (30–150 nm in diameter), which act as paracrine mediators between MSCs and target cells [99,100,101]. Exosomes can deliver cargo (nucleic acids, proteins, lipids, amino acids, and metabolites) from the originating cells to the target cells [101,102,103]. Compared with those of artificial nanocarriers, exosomes, as natural vesicles secreted by cells, have double lipid membranes, better biocompatibility, lower immunogenicity, stronger targeting specificity, deeper tissue permeability, and longer circulating half-life [101,104,105,106,107]. Based on these advantages, exosomes have been applied for engineering functional cargo loads, such as packaged nucleic acid, functional proteins, and other therapeutic molecules into exosomes [100,101,108,109,110]. MSC-derived exosomes have been recently transfected with functional RNAs to target cells, suggesting their potential as an alternative for cell-based therapy.

Notably, exosomes have been shown to transfer microRNAs (miRNAs) to target cancer cell proliferation, differentiation, and metastasis [109,111,112,113,114]. For example, miRNA-100 carried by MSC-derived exosomes suppress tumor angiogenesis and breast cancer progression via the mTOR/HIF1A/VEGF pathway [115]. Additionally, Li et al. showed that engineered exosomes from UCMSCs enriched with miR-302a significantly inhibit endometrial cancer cell proliferation and migration by decreasing cyclin D1 expression and inhibiting the AKT pathway [116]. Furthermore, modified miR-199a derived from AD-MSC exosomes can improve hepatocellular carcinoma (HCC) chemosensitivity through the mTOR signaling pathway [117]. Therefore, these observations suggest that exogenous miRNA delivered by MSC-derived exosomes could be an effective anticancer therapy strategy.

On modifying the normal fibroblast-like mesenchymal cell-derived exosomes to deliver short interfering RNA or short hairpin RNA to target oncogenic *KrasG12D*, tumor growth was found to be decreased in multiple mouse models of pancreatic cancer, thereby increasing the overall survival [118]. This study offers insight into the target therapeutic potential of exosomes in pancreatic cancer. Furthermore, infusing the exosomes derived from miRNA-engineered MSCs, which contain abundant MSC-sourced anti-tumorigenic miRNAs, can represent a potentially new therapeutic measure for pancreatic cancer. For example, the exosomes extracted from miR-1231-modified BM-MSCs with high levels of miR-1231 reduce the proliferation of pancreatic cancer cells [119]. Overexpressed miR-126-3p derived from BM-MSC exosomes inhibit the proliferation, invasion, and metastasis of pancreatic tumor cells and induce their apoptosis in vitro and in vivo by inhibiting the expression of ADAM9 [120]. Exosomes derived from miRNA-MSCs release miR-124 and miR143-3p in pancreatic tumor cells, inhibiting the proliferation of tumor cells [121,122]. The exosomes derived from miRNA-128-3p-modified UCMSCs can inhibit the proliferation, invasion, and migration of pancreatic cancer cells via the miRNA-128-3p/Galectin-3 axis [123]. Yao et al. found that BM-MSC-derived exosomes reduce the expression of miR-338-5p in pancreatic cancer cells via circ_0030167 and targeting WIF1, thereby inhibiting the Wnt8/β-catenin signaling pathway [124]. Notably, in a recent study, the exosomes were applied for cargo loading miR-145-5p. Exosomes derived from UCMSCs modified with an intra-tumor injection of miR-145-5p, which deliver exogenous miR-145-5p, decrease the growth of pancreatic tumors by inhibiting the expression of SMAD3 [125]. This study revealed a novel insight that exosomes might be an attractive therapeutic tool for the clinical administration of miRNAs in patients with PDAC. Taken together, exogenous functional RNAs released by exosomes derived from MSCs may serve as potential targets for treating pancreatic cancer (Figure 1).

### 4.4. MSC-Mediated Drug Delivery

MSCs can incorporate chemotherapeutic drugs in vitro, subsequently releasing the effective concentration of drugs in their conditioned medium to exert therapeutic effects [126,127,128]. Furthermore, tumor-homing properties of MSCs allow them to precisely deliver the drug to the tumor location; this has been widely studied as a targeted delivery agent of anti-cancer drugs [27,78,129]. For example, paclitaxel (PTX)-loaded MSCs inhibit the growth of leukemia cells, decrease angiogenesis, and increase survival [130]. Moreover, PTX-loaded MSCs inhibit the proliferation of human myeloma cells [131]. 

The proliferation of stromal fibroblasts and the deposition of extracellular matrix, which are the defining characteristics of PDAC, lead to a fibrotic state known as desmoplastic or reactive stroma [45,132]. Therefore, this could make it difficult to acquire an effective drug concentration by the common route of administration. Notably, after the MSCs were preconditioned to high doses of PTX, they intracellularly accumulate the drug and then release it, thereby inhibiting pancreatic tumor cell proliferation [133]. Brini et al. demonstrated that PTX-loaded gingival interdental papilla MSCs can release a sufficient amount of PTX to inhibit the proliferation of pancreatic tumor cells [134]. Moreover, gemcitabine (GCB)-loaded BM-MSCs decrease the proliferation of human pancreatic cancer cells in vitro [135]. Nevertheless, PTX and GCB have not yet been shown to be delivered in the PDAC location by MSCs. Thus, MSC-mediated delivery of anti-cancer drugs needs to be further explored in vivo.

Exosomes can be loaded with therapeutic drugs and then used to release them into the target cells [136,137]. The methods applied for directly loading drugs inside exosomes include incubation, electroporation, sonication, extrusion, freeze-thaw cycles, and saponin application. Presently, the most commonly applied methods are incubation and electroporation [114,137]. As drug carriers, exosomes are widely studied as therapeutic agents and can potentially be clinically applied. Recently, GEMP- and PTX-loaded exosomes revealed superiorities in homing and penetrating abilities that aided in inhibiting the growth of pancreatic tumors in vivo [138]. Additionally, Zhou et al. found that BM-MSC-derived exosomes containing electroporation-loaded galectin-9 siRNA and modified with oxaliplatin (OXA) increase drug accumulation in the tumor location and reduce the growth of pancreatic tumors [139]. Thus, MSC-derived exosomes could act as a potential nanoscale drug delivery platform for treating pancreatic cancer (Figure 1).

### 4.5. Delivery of Oncolytic Viruses 

The oncolytic virus has revealed promising results in the treatment of several cancers in various clinical trials [140,141,142]. It can directly cause oncolysis and spread to adjacent tumor cells to activate an anti-cancer immune response. Oncolytic viruses can replicate and selectively target tumor cells, but they cannot bind or effectively replicate in most normal cells. MSCs have been shown to protect viruses from immune clearance through a unique cell carrier tool before delivering them to metastatic tumor sites [143,144,145]. Although the tumor-homing ability of MSCs makes them a promising candidate for systemically delivering oncolytic viruses to tumor location, infection and particle production by MSCs remain areas of concern. The viruses genetically modified for improved delivery by MSCs are aimed at enhancing oncolysis and improving virus production in tumor cells [146].

Kaczorowski et al. showed that oncolytic adenovirus-TRAIL-modified BM-MSCs can precisely migrate to tumor sites, but not in normal cells, inhibiting the growth of pancreatic cancer [147]. Na et al. found that oncolytic adenovirus-RLX-PCDP-loaded BM-MSC carrier can be released to target pancreatic tumors and induce effective viral replication and relaxin expression, which inhibit the growth of pancreatic cancer [148]. Additionally, myxoma-TNFSF14-loaded AD-MSCs can home to pancreatic cancer sites; oncolytic viruses are efficiently delivered, inhibit the growth of PDAC, and increase overall survival [57]. Altogether, oncolytic virus-mediated gene treatment is a promising approach to treating pancreatic cancer (Figure 1). However, some drawbacks of using a virus as a vector for clinical therapy, including packaged space limitation, increased risk of insertional mutagenesis, immune clearance, and limitations of producing high titer virus for clinical application, need to be overcome. 

## 5. Challenges of MSCs in Treating Pancreatic Cancer 

MSCs have been used as a therapeutic intervention for tumors; nevertheless, they are reportedly involved in tumor progression, including tumorigenesis, tumor growth, metastasis, and regulation of the TME [23,113]. Furthermore, the potential mechanisms by which MSCs crosstalk with tumor cells in the TME have not yet been elucidated [22,23,24,149]. Hence, the clinical application of MSCs in the treatment of pancreatic cancer remains controversial and challenging (Figure 2). 

### 5.1. Tumorigenicity

Numerous studies have shown that MSCs have inherent tumorigenicity properties [17,55,150,151,152,153]. MSCs possess the molecular potential to affect and direct several crucial processes, which are important for tumor development, as the cells contain an abundant source of various biochemical mediators [14,22]. MSCs have been successfully isolated from various types of tumor tissues, such as HCC, glioma, gastric cancer, breast cancer, ovarian cancer, prostate cancer, colon cancer, and pancreatic cancer, indicating that MSC is a distinct stromal cell type in the TME that participates in tumor development [17,154]. It consists of stromal cells that include tumor-associated fibroblasts, tumor endothelial cells, immune and inflammatory cells, and bone marrow-derived cells [155,156]. Interactions between tumor cells and the TME tremendously impact tumor development, metastasis, and drug resistance [157,158]. 

MSCs can reportedly modulate stromal heterogeneity in various solid tumors, including pancreatic cancer [151,159]. Furthermore, MSCs regulate specific secretory molecules in the TME and promote the progression and invasion of pancreatic cancer [152,160]. AD-MSCs migrate to pancreatic cancer locations to serve as a major source of a-SMA+ cells and promote tumor progression [55,56]. Nevertheless, the mechanism underlying the mobilization of these intricate molecules remains unclear. Notably, Ganguly et al. found that MUC5AC acts as a systemic carrier of tumor secretome and can alter stromal maturation in pancreatic cancer by mobilizing AD-MSCs via CD44 and CD29/ITGB1 clustering [161]. Moreover, Miyazaki et al. found that AD-MSCs can differentiate into different pancreatic cancer-associated fibroblast subtypes, thus driving tumor heterogeneity and playing a key role in the development and drug resistance of PDAC [162]. Kabashima et al. found that BM-MSC-derived myofibroblasts modulate epithelial-to-mesenchymal transition and augment stemness-associated genes in PDAC [163]. Furthermore, BM-MSCs can reportedly migrate to tumor vessels and promote pancreatic cancer angiogenesis [164]. 

Tumor-associated macrophages (TAMs) are the predominant immune cells in the TME and are involved in tumorigenesis, immune escape, metastasis, and chemotherapeutic resistance [165]. TAMs have become a potential target for developing new cancer treatments [166] and are frequently involved in pancreatic cancer progression and the Warburg effect [167]. Reportedly, MSCs promote the progression and growth of pancreatic cancer by inducing alternating polarization of macrophages [168]. This could provide a potential new therapeutic strategy for PDAC.

Moreover, UCMSC-derived exosomes promote the growth of PDAC, which might be regulated via the delivery of miRNAs to the tumor cells to mediate the relevant signaling pathways [169]. Thus, MSCs promote the progression of pancreatic cancer.

### 5.2. MSCs Promote Drug Resistance

MSCs have been revealed to induce and play an important role in the drug resistance of tumor cells in the TME [136,170]. Several potential mechanisms underlying this phenomenon might include promoting active drug sequestration, decreasing drug concentration, and delivering specific RNA, proteins, and functional small molecules into target cells to induce dysregulation of relevant signaling pathways. For example, Roodhart et al. found that endogenous MSCs are activated on treatment with platinum analogs and release some mediators to protect tumor cells against a range of chemotherapeutics. By a metabolomics method, the results showed that two distinct platinum-induced polyunsaturated fatty acids derived from MSCs, 12-oxo-5,8,10-heptadecatrienoic acid and hexadeca-4,7,10,13-tetraenoic acid [16:4 (n−3)], induce resistance to platinum-based chemotherapy [171]. Furthermore, MSC-derived exosomes induce drug resistance in various tumor cells. Wang et al. showed that BM-MSC-derived exosomes play key roles in drug resistance in multiple myeloma and induce their proliferation, migration, and survival [172]. Moreover, MSC-derived exosomes promote drug resistance in gastric cancer cells via activating the CaM-Ks/Raf/MEK/ERK pathway [173]. MSCs promote the expression of SNHG7 in pancreatic tumor cells, inducing the stemness and FOLFIRINOX resistance through the Notch1/Jagged1/Hes-1 axis [174]. Hence, MSCs might induce drug resistance in pancreatic tumor cells; these obstacles should be overcome for MSCs to be clinically applied.

### 5.3. Limitations of the Clinical Applications of MSCs

Several studies have demonstrated the efficacy and safety of various types of MSCs and MSC-derived exosomes in pancreatic cancer therapy; nevertheless, their clinical applications have some limitations.

#### 5.3.1. Routines of MSC Application and Distribution in the Host

Numerous studies have reported the tumor-homing properties of MSCs; however, the migration and distribution of MSCs in the body are not yet clearly understood. In animal experiments, the most common method of infusion of MSCs is through the intravenous (IV) or intraperitoneal (i.p.) route [175]. Nevertheless, owing to their size and the small dimensions of the lung vessels, a large number of MSCs are temporarily distributed in the lung after IV administration [176,177,178,179,180]. Notably, three days after IV administration, most of the MSCs are recruited to the tumor locations in an orthotopic pancreatic cancer model of athymic nude mice; only some MSCs are observed in the lung [164]. An orthotopic xenograft model of human pancreatic cancer built by directly injecting Panc02 pancreatic cancer cells into the pancreas and followed by intravenously injecting MSCs showed that three days after administration, the MSCs are recruited to the growing tumor vasculature of cancer [95]. Similarly, when MSCs are intraperitoneally injected into an orthotopic xenograft model of human pancreatic cancer built using i.p. PANC-1, most MSCs are observed in specific organ sites of the tumor and metastatic lesions after seven days [96]. After three weeks, MSCs are observed only in the tumor tissues but not in the fat tissues or on the intraperitoneal organ surfaces [76]. These results indicate that intravenously and intraperitoneally administered MSCs can primarily migrate to the pancreatic cancer site, thus inhibiting the growth of tumors. Upon injecting pancreatic cancer cells into the pancreatic nape of the neck and simultaneously orthotopically injecting MSCs into the pancreas, a higher pancreas-targeted distribution of the MSCs is noticed than that observed when MSCs are intravenously injected [57]. Therefore, the intravenously, intratumorally, and orthotopically infused MSCs may be effective; nonetheless, determining the best route of administration requires further investigation.

#### 5.3.2. Timing and Dosage

The anti-tumor effect of exogenous MSCs depends on the time of their inoculation in tumor-bearing animals. MSCs infused during the initial phase of tumor growth exert anti-tumor effects, whereas those infused during the progressive stage of tumor development induce immune escape and promote the growth of the tumor [181]. Thus, the best time for treatment needs to be established. According to current studies, when MSCs are infused in the pancreatic cancer model during the early stage of solid tumorigenesis, the growth of tumors can be inhibited [57,76,97]. However, infusing MSCs in the pancreatic cancer model during the progressive stage of tumor development (10–17 days) can also inhibit the growth of tumors [75,89,92,95]. Although approximately 1 × 10^5^ to 1 × 10^6^ cells are usually infused in pancreatic cancer animal models, the most appropriate dose of MSCs has not yet been confirmed [57,89,94,97]. Therefore, the optimal dose and route of administration need to be elucidated.

#### 5.3.3. Infiltration, Persistence, and Exhaustion

Although CAR-modified T cells reveal a new potential option in cancers, current CAR T-cell clinical trials in pancreatic cancer have been unable to increase survival and exhibit any significant response. This might be due to obstacles such as T-cell infiltration, persistence, and exhaustion [45]. The TME in pancreatic cancer comprises tumor cells, endothelial cells, immune cells, stromal cells, extracellular matrix, and a broad spectrum of enzymes, cytokines, and growth factors [182]. As a dense stroma surrounds pancreas cancer, successful immunotherapy and chemotherapy are required to break through the physical and environmental barriers in the TME [183]. Additionally, owing to the complex TME, CAR T-cell treatment in pancreatic cancer leads to T-cell exhaustion and persistence [184,185]. Similarly, successful MSCs treatments for pancreatic cancer should also be able to overcome the physical and environmental barriers in the TME, including infiltration, persistence, and exhaustion. Thus, further clinical trials need to be carried out.

MSC isolation and expansion remain a challenge. Even when subjected to similar isolation and culture conditions, the MSCs obtained from different sources can be heterogeneous [186]. This observation reveals the heterogeneous nature of a typical MSC population and their potential to migrate to different tissues, promote tissue repair, or inhibit inflammation [187]. Thus, the differences in approaches used for culturing and expanding MSCs need to be studied to determine whether these differences affect their phenotype and functional properties [187]. Furthermore, standard protocols for characterizing MSC-derived exosomes owing to their heterogeneous nature that may have diverse effects on the target cells remain lacking [188]. Moreover, patients who receive MSC-derived exosomes need to be closely monitored to determine the optimal dosage with the best therapeutic efficiency [109]. Lastly, MSCs manufactured for cancer therapy should meet the standards of good manufacturing practices and regulations.

## 6. Conclusions and Perspectives

MSCs can home to tumor locations and survive in the TME owing to their tumor-homing properties. Thus, MSCs or MSC-derived exosomes, as a carrier of anti-cancer drugs, can be genetically modified to deliver various agents to inhibit tumor growth. The application of MSCs or MSC-derived exosomes as carriers for tumor target therapy has numerous advantages, including low immunogenicity, tumor tropism, easy rapid isolation and expansion, and the ability to release various therapeutic agents. In the recent decade, remarkable progress has been made in the field of engineered MSC-based tumor-targeted therapy for pancreatic cancer. However, the clinical application of MSC-based therapy in the treatment of pancreatic cancer still faces many challenges. Thus, overcoming these challenges is necessary. Subsequently, the crosstalk between MSCs and tumor cells to increase the clinical safety of MSC-based therapeutic measures need to be clarified. Thus, future research should be focused on the long-term follow-up of MSC-treated tumor-bearing animals to address all safety concerns related to the plasticity of MSCs and their possible pro-tumorigenic effects. In summary, MSC-based therapies are emerging as an attractive option for the treatment of pancreatic cancer.

## Figures and Tables

**Figure 1 ijms-24-03559-f001:**
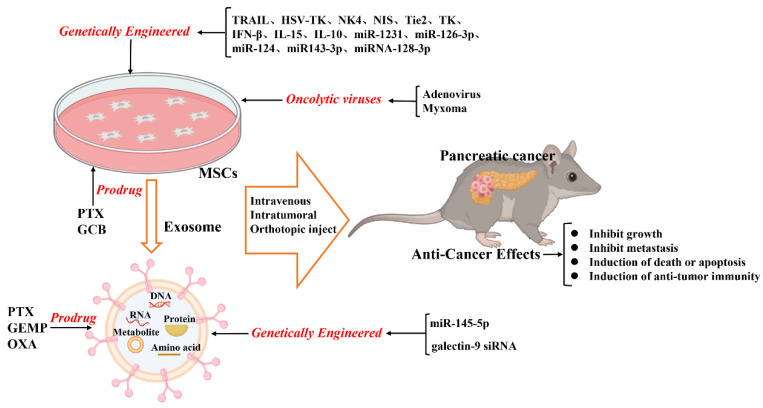
Schematic illustration of the current anticancer treatment based on mesenchymal stromal cells (MSCs) and MSC-derived exosomes.

**Figure 2 ijms-24-03559-f002:**
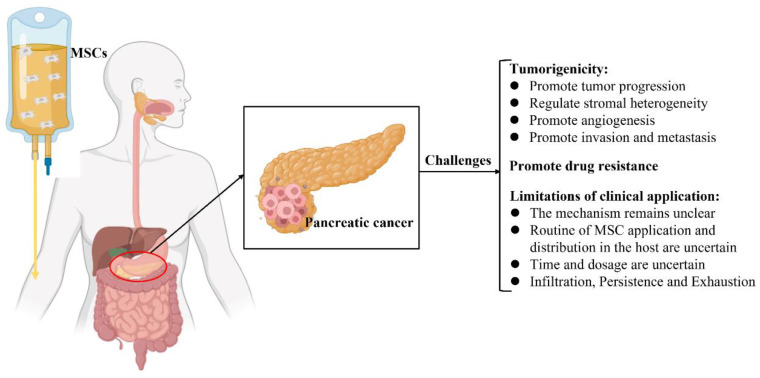
Schematic illustration of the current challenges of mesenchymal stromal cell treatment in pancreatic cancer.

## Data Availability

Not applicable.

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
