# Peer review of "Mesenchymal Stromal Cell-Based Targeted Therapy Pancreatic Cancer: Progress and Challenges"

_ijms, 2023, doi:10.3390/ijms24043559_

Round 1

Reviewer 1 Report

Ma et al. summarized the current progress of the applications of mesenchymal stromal cells (MSC) in the treatment of pancreatic cancers and also thoroughly discussed the limitations and challenges of MSC-based therapies.  Overall, this is a well-organized manuscript. I believe it will have broad readers and will definitely benefit both basic scientific research and clinical application for the treatment of pancreatic cancers. I would recommend publishing this Review in the International Journal of Molecular Sciences.

But before acceptance, the authors would check and polish their writings. Some obvious grammar errors exist both in the text and in the Figures from the beginning to the end of the manuscript.

Author Response

Point 1: Ma et al. summarized the current progress of the applications of mesenchymal stromal cells (MSC) in the treatment of pancreatic cancers and also thoroughly discussed the limitations and challenges of MSC-based therapies. Overall, this is a well-organized manuscript. I believe it will have broad readers and will definitely benefit both basic scientific research and clinical application for the treatment of pancreatic cancers. I would recommend publishing this Review in the International Journal of Molecular Sciences. But before acceptance, the authors would check and polish their writings. Some obvious grammar errors exist both in the text and in the Figures from the beginning to the end of the manuscript.

Response 1: Thanks for the constructive suggestions to improve our manuscript. And sorry for the language problem, we have now woked on language and have also involved native English speakers for language corrections. We really hope that the flow and language level have been substantially improved.

Reviewer 2 Report

This review nicely covers advantages and challenges in MSC therapy for pancreatic cancer. Below are my suggestions/comments. 

  • Page 3, correct abreviviationfor Tumor Microenvironment is "TME" not TEM. This sould be corrected on Page 19, abbreviation section too.
  • Tumor-Associated Macrophages (TAMs) are predominant immune cells in tumor microenvironemnt and are invloved intumorigenesis, immune escape, metastasis, and chemotherapeutic resistance. On the otherhand, MSCs are known to modulate TAMs. Although authors brifly mentioned about macrophage polarization in develpoment of pancreatic cancer under section 5.1 (page 15 ), I think this section may be expanded by including studies focusing on role of TAMs in pancreatic cencer and/or MSC-dependent TAMs modulation.

Author Response

Point: This review nicely covers advantages and challenges in MSC therapy for pancreatic cancer. Below are my suggestions/comments. 

Page 3, correct abbreviation for Tumor Microenvironment is "TME" not TEM. This should be corrected on Page 19, abbreviation section too.

Tumor-Associated Macrophages (TAMs) are predominant immune cells in tumor microenvironment and are involved in tumorigenesis, immune escape, metastasis, and chemotherapeutic resistance. On the other hand, MSCs are known to modulate TAMs. Although authors briefly mentioned about macrophage polarization in development of pancreatic cancer under section 5.1 (page 15), I think this section may be expanded by including studies focusing on role of TAMs in pancreatic cancer and/or MSC-dependent TAMs modulation.

Response:

1. Thanks for the constructive suggestions to improve our manuscript, we have corrected abbreviation for Tumor Microenvironment (TME) in the Page 3 and Page 19.

2. Thank you very much for reviewing our paper carefully and giving us lots of detailed suggestions. According to your suggestion, we have expanded some contents by including studies focusing on role of TAMs in pancreatic cancer and/or MSC-dependent TAMs modulation.

Reviewer 3 Report

1.       The purpose of the review is unclear; in the abstract, it does not coincide with that indicated in the introduction. The conclusion also does not reflect the idea for which the review was written.

2.       Among the literature sources, reviews predominate. This reduces the value of claims, for example after the statement "Numerous studies have shown that MSCs have inherent tumorigenicity properties" all 3 references are reviews and none of the original studies.

3.       In many sections, it is not obvious which MSCs are referred to - introduced allogeneic, own normal stromal progenitor cells of the body or isolated from the tumor microenvironment.

4.       Cells are either multipotent or stem, one of the two. Mesenchymal stem cells and multipotent mesenchymal stromal cells are not the same (Dominici et al., 2006).

5.       If the ability of MSCs to migrate to the tumor area has been confirmed in vivo, then references should be provided.

6.       It is not clear why the tumor microenvironment containing blood vessels is a physical barrier to the penetration of chemotherapy drugs. The statement about extreme lack of neovascularization requires additional reference.

7.       What idea underlies the genetic modification of MSCs and their subsequent intratumoral injection? If for the sake of the secretion of factors, then how is this better than the direct introduction of the factors themselves?

8.       MSCs vary greatly in their characteristics and composition of secreted exosomes, factors, and molecules, depending on the source and donor. Why are MSC exosomes so attractive for microRNA packaging?

9.       The word "patients" is usually applied to humans, it should not be used in relation to animals (For example, paclitaxel (PTX)-loaded MSCs inhibit the growth of leukemia cells, decrease angiogenesis, and increase the survival of patients with leukemia (in reference " improve survival of leukaemia-bearing mice")).

Author Response

Point 1: The purpose of the review is unclear; in the abstract, it does not coincide with that indicated in the introduction. The conclusion also does not reflect the idea for which the review was written.

Response 1: Thank you very much for reviewing our paper carefully and giving us lots of detailed suggestions. According to your suggestion, we have substantially modified the abstract/introduction/conclusion to better present these novel points of our work.

Point 2: Among the literature sources, reviews predominate. This reduces the value of claims, for example after the statement "Numerous studies have shown that MSCs have inherent tumorigenicity properties" all 3 references are reviews and none of the original studies.

Response 2: Thank you for your suggestion, we have revised it in the manuscript.

Point 3: In many sections, it is not obvious which MSCs are referred to - introduced allogeneic, own normal stromal progenitor cells of the body or isolated from the tumor microenvironment.

Response 3: In accordance with your suggestion, we have introduced the source of MSCs in detail in each section.

Point 4: Cells are either multipotent or stem, one of the two. Mesenchymal stem cells and multipotent mesenchymal stromal cells are not the same (Dominici et al., 2006).

Response 4: We agree with you that Mesenchymal stem cells and multipotent mesenchymal stromal cells are not the same. Accordingly, we have corrected this error in the manuscript.

Point 5: If the ability of MSCs to migrate to the tumor area has been confirmed in vivo, then references should be provided.

Response 5: In accordance with your suggestion, we have provided references in the manuscript (J. Houghton, C. Stoicov, S. Nomura, A.B. Rogers, J. Carlson, H. Li, X. Cai, J.G. Fox, J.R. Goldenring, T.C. Wang, Gastric cancer originating from bone marrow-derived cells, Science, 306 (2004) 1568-1571; Y. Zhang, A. Daquinag, D.O. Traktuev, F. Amaya-Manzanares, P.J. Simmons, K.L. March, R. Pasqualini, W. Arap, M.G. Kolonin, White adipose tissue cells are recruited by experimental tumors and promote cancer progression in mouse models, Cancer Res, 69 (2009) 5259-5266; T. Okumura, K. Ohuchida, S. Kibe, C. Iwamoto, Y. Ando, S. Takesue, H. Nakayama, T. Abe, S. Endo, K. Koikawa, M. Sada, K. Horioka, N. Mochidome, M. Arita, T. Moriyama, K. Nakata, Y. Miyasaka, T. Ohtsuka, K. Mizumoto, Y. Oda, M. Hashizume, M. Nakamura, Adipose tissue-derived stromal cells are sources of cancer-associated fibroblasts and enhance tumor progression by dense collagen matrix, Int J Cancer, 144 (2019) 1401-1413; J. Jazowiecka-Rakus, A. Hadrys, M.M. Rahman, G. McFadden, W. Fidyk, E. Chmielik, M. Pazdzior, M. Grajek, V. Kozik, A. Sochanik, Myxoma Virus Expressing LIGHT (TNFSF14) Pre-Loaded into Adipose-Derived Mesenchymal Stem Cells Is Effective Treatment for Murine Pancreatic Adenocarcinoma, Cancers (Basel), 13 (2021); J. Han, H.S. Hwang, K. Na, TRAIL-secreting human mesenchymal stem cells engineered by a non-viral vector and photochemical internalization for pancreatic cancer gene therapy, Biomaterials, 182 (2018) 259-268).

Point 6: It is not clear why the tumor microenvironment containing blood vessels is a physical barrier to the penetration of chemotherapy drugs. The statement about extreme lack of neovascularization requires additional reference.

Response 6: We agree with you that is not clear why the tumor microenvironment containing blood vessels is a physical barrier to the penetration of chemotherapy drugs. Accordingly, we have corrected this error in the manuscript.

Point 7: What idea underlies the genetic modification of MSCs and their subsequent intratumoral injection? If for the sake of the secretion of factors, then how is this better than the direct introduction of the factors themselves?

Response 7:

We have consulted lots of literatures for replying your interesting questions, there are few articles report that only MSCs or genetically modified MSCs are directly injected into tumor tissue. However, recently, some articles have reported that MSCs carrying the oncolytic viruses be directly injected into the tumor tissue and play a key role in anti-cancer. The oncolytic virus can directly oncolysis and spread to adjacent tumor cells to activate anti-cancer immune response. Oncolytic viruses have the ability to replicate and selectively target tumor cells, but they cannot bind or replicate effectively in most normal cells. MSCs have been shown to protect viruses from immune clearance through a unique cell carrier tool before delivering them to metastatic tumor sites. The viruses modified genetically for improved delivery by MSCs aim to enhance oncolysis and improve virus production in tumor cells.

Point 8: MSCs vary greatly in their characteristics and composition of secreted exosomes, factors, and molecules, depending on the source and donor. Why are MSC exosomes so attractive for microRNA packaging?

Response 8: Although MSCs have good therapeutic effects, there are also several limitations, including difficulty in generating a consistent source of cells with a stable phenotype and maintenance of biological activity, infusion toxicities caused by large cells physically trapped in the lung microvasculature, ectopic tissue formation, tumorigenicity, quantification of bioactive substances, and the logistics of delivery. MSC-based treatment effects can be attributed to the MSC-sourced secretomes, which consist of a soluble component and encapsulated microvesicles and exosomes. Compared with artificial nano carriers, exosomes, as natural vesicles secreted by cells, have double lipid membranes, better biocompatibility, lower immunogenicity, stronger targeting specificity, deeper tissue permeability and longer cycle half-life. Based on these advantages, exosomes have been applied for engineering functional cargo loads, such as the package of nucleic acid, functional proteins, and other therapeutic molecules into exosomes.

Point 9: The word "patients" is usually applied to humans, it should not be used in relation to animals (For example, paclitaxel (PTX)-loaded MSCs inhibit the growth of leukemia cells, decrease angiogenesis, and increase the survival of patients with leukemia (in reference " improve survival of leukaemia-bearing mice")).

Response 9: Thanks for the constructive suggestions to improve our manuscript, we have corrected this error in the manuscript.

Round 2

Reviewer 3 Report

Comments have been positively addressed by authors.